# Immunology of THymectomy And childhood CArdiac transplant (ITHACA): protocol for a UK-wide prospective observational cohort study to identify immunological risk factors of post-transplant lymphoproliferative disease (PTLD) in thymectomised children

Ugonna T Offor ⬥ ,[1,2] Paolo Hollis,[3] Milos Ognjanovic,[4] Gareth Parry,[5] Abbas Khushnood,[5] Heather M Long,[6] Andrew R Gennery,[7,8] Chris M Bacon,[1,9] Jacob Simmonds,[3] Zdenka Reinhardt,[5] Simon Bomken[1,2]

For numbered affiliations see end of article.

**Correspondence to**
Dr Ugonna T Offor;
ugo.offor@newcastle.ac.uk

## ABSTRACT

**Introduction**  Paediatric heart transplant patients are disproportionately affected by Epstein-Barr virus (EBV)-related post-transplant lymphoproliferative disease (PTLD) compared with other childhood solid organ recipients. The drivers for this disparity remain poorly understood. A potential risk factor within this cohort is the routine surgical removal of the thymus—a gland critical for the normal development of T-lymphocyte-mediated antiviral immunity—in early life, which does not occur in other solid organ transplant recipients. Our study aims to describe the key immunological differences associated with early thymectomy, its impact on the temporal immune response to EBV infection and subsequent risk of PTLD.

**Methods and analysis**  Prospective and sequential immune monitoring will be performed for 34 heart transplant recipients and 6 renal transplant patients (aged 0–18 years), stratified into early (<1 year), late (>1 year) and non-thymectomy groups. Peripheral blood samples and clinical data will be taken before transplant and at 3, 6, 12 and 24 months post-transplant. Single cell analysis of circulating immune cells and enumeration of EBV-specific T-lymphocytes will be performed using high-dimensional spectral flow cytometry with peptide-Major Histocompatibilty Complex (pMHC) I/II tetramer assay, respectively. The functional status of EBV-specific T-lymphocytes, along with EBV antibodies and viral load will be monitored at each of the predefined study time points.

**Ethics and dissemination**  Ethical approval for this study has been obtained from the North of Scotland Research Ethics Committee. The results will be disseminated through publications in peer-reviewed journals, presentations at scientific conferences and patient-centred forums, including social media.

**Trial registration number**  ISRCTN10096625.

## STRENGTHS AND LIMITATIONS OF THIS STUDY

⇒ This is the first prospective study to monitor the temporal immune response to Epstein-Barr virus (EBV) infection within a group of patients known to be at a high risk of developing post-transplant lymphoproliferative disease.

⇒ The integration of data from clinical parameters, in-depth immunophenotyping and EBV-specific T-lymphocyte functional assay permits robust analysis of potential predictive immune biomarkers.

⇒ The multicentre study design optimises recruitment of participants.

⇒ The primary study limitation is the low incidence of paediatric heart transplantation, which may affect the rate of study recruitment.

## INTRODUCTION

Post-transplant lymphoproliferative disease (PTLD) is the most common childhood cancer in paediatric recipients of a solid organ transplant. This heterogeneous group of life-threatening lymphoid malignancies is typically driven by Epstein-Barr virus (EBV) infection.[1] PTLD comprises a histological spectrum that ranges from indolent non-destructive B-cell lymphoid infiltrates through to more aggressive and destructive polyclonal or monoclonal B-cell lymphomas, for example, Burkitt lymphoma and diffuse large B-cell lymphoma. These destructive PTLD subtypes are indistinguishable from high-grade mature B-cell lymphomas seen



in immunocompetent children.[2] Their incidence is allograft-dependent and ranges from 1%–2% in renal transplant recipients to as high as 10%–15% in heart transplant patients.[1 3]

PTLD has one of the worst clinical outcomes among childhood lymphomas. The estimated 2-year event-free survival (EFS) is 70% compared with 94% in sporadic cases within the general paediatric population.[4–6] In even starker contrast is the survival outcome for paediatric heart transplant patients, who ostensibly have an inferior EFS compared with other non-cardiac organ transplants.[7 8] The reasons for this potential disparity are still poorly understood, although partly ascribed to a higher incidence of therapeutic complications from organ rejection and treatment-related toxicities.[9]

The pathogenesis of PTLD is complex and multifactorial. It involves the interplay between EBV-driven lymphoproliferation, iatrogenic immunosuppression and the suspected functional exhaustion of T-lymphocytes due to graft-initiated chronic antigen stimulation.[10] The role of EBV is clearly established in children, many of whom experience primary infection from an EBV-mismatched organ.[1 3 4] In vitro and in vivo studies focused mainly on adult monomorphic PTLD have demonstrated distinct patterns of viral protein expression in infected B-lymphocytes.[10 11] These expression patterns likely influence both the cellular composition of the tumour immune microenvironment and aberrant immune signalling, which result in immune escape of tumour cells.[10]

In paediatric heart transplant patients, cardiac surgery via median sternotomy in early childhood often requires routine thymectomy in order to access the heart and great vessels. Our earlier study retrospectively examined risk factors for PTLD in the largest UK cohort of paediatric orthotopic heart transplant patients to date.[3] The risk of PTLD was found to be significantly higher in children with congenital heart disease (CHD) (HR=3.2; 95% CI=1.4 to 7.4) and early thymectomy in infancy (HR=2.7; 95% CI=1.3 to 5.2).[3] Furthermore, children with CHD had persistently lower T-lymphocyte counts compared with children transplanted for acquired cardiomyopathy (CD4+: 430 cell/µL vs 963 cells/µL, p<0.01 and CD8+: 367 cells/µL vs 765 cells/µL, p<0.01).[3] Similar studies have identified marked phenotypic and functional disruptions to the T-lymphocyte compartment in paediatric heart transplant patients compared with children receiving a liver or kidney transplant.[12 13]

During the first year of life, the thymus plays a crucial role in the development of cell-mediated immunity, providing a microenvironment for precursor T-lymphocytes to proliferate and differentiate into mature (naïve) T-lymphocytes.[14] While it has been shown that neonatal thymectomy alone is associated with premature immunosenescence and an increased risk of viral infections such as Cytomegalovirus (CMV),[14] little is known about the immunological consequences of early thymectomy in immunosuppressed transplant patients. In particular, there are no data on the synergetic impact of thymectomy and transplant-related immunosuppression on EBV-specific immunity and the subsequent risk of developing PTLD.

### Study aims and hypothesis

This study aims to investigate the development of EBV-specific immune responses following childhood heart transplant. Specifically, it aims to identify the impact of early thymectomy—compounded by iatrogenic immunosuppression—on EBV immunology and the risk of PTLD.

We hypothesise that a combination of early thymectomy and lifelong immunosuppression therapy establishes a tolerogenic immune profile consisting of dysregulated, exhausted and senescent immune cell subsets, poorly able to control EBV infection. This dysfunctional immune microenvironment permits the uncontrolled proliferation of EBV-infected B-lymphocytes and the subsequent development of PTLD.

### METHODS AND ANALYSIS
### Study design and setting

The Immunology of THymectomy And Childhood CArdiac transplant (ITHACA) study is a prospective nationwide cohort study recruiting children (0–18 years) from the two UK centres currently commissioned to provide paediatric heart transplant services: The Freeman Hospital, Newcastle upon Tyne and Great Ormond Street Hospital for Children, London. The study cohort will consist of 34 prospective heart transplant recipients and a non-thymectomy age-matched control group made up of 6 renal transplant recipients. All patients meeting the study's eligibility criteria will be identified and recruited through their local transplant teams at the time of listing for cardiac or renal transplantation. The study opened to patient recruitment in March 2022 and is expected to recruit until June 2024.

### Inclusion criteria

► Aged 0–18 years.
► Actively listed on the National Health Service Blood and Transplant (NHSBT) waiting list for a primary organ transplant or awaiting transplant with a living related donor kidney or recently transplanted with pretransplant blood samples available.
► Written informed consent.

### Exclusion criteria

► Has a pre-existing diagnosis of an inherited or acquired immunodeficiency.
► Has an underlying thymic disorder.
► Has previously received a bone marrow or organ transplant.
► Has had a previous cancer diagnosis.
► Withheld consent.
► Weight under 2.5 kg.

### Informed consent

Informed consent will be obtained from the parent/carer of the eligible child or from the patient themselves if over

the age of 16 years. Assent may be given by children<16 years of age who wish to do so.

A deferred consent approach will be employed for potential study participants who attend a research site's transplant service for transplantation in a critical/ life-threatening clinical situation. This will involve the collection of baseline study samples from potential participants at the point of presurgical workup without written informed consent being received. Discussion about the study, the giving of participant information sheet (PIS) and receiving of written informed consent/assent will be offered at a more appropriate time before further follow-up blood tests are taken. Such cases will require that the clinical team considers obtaining informed consent prior to transplant to be inappropriate. This will be documented in the patient's clinical notes. Study samples collected under such circumstances will be processed for storage but not analysed until written informed consent is obtained. Any patient who has study samples collected by deferred consent but subsequently declines enrolment in the study will have their samples destroyed in a timely manner according to local laboratory standard operating procedures. Additional consent will be sought to store specimens for future ethically approved research.

### Assessment and procedures

Children will undergo blood sampling for study-specific investigations during routine clinical visits prior to transplant and at 3, 6, 12 and 24 months post-transplant (figure 1). No additional study-related hospital visits are planned. Medical therapy will be applied as clinically indicated and per the local post-transplantation protocols. A total of 5–20 mL of blood will be taken at each study time point for all participants. This will coincide with their usual transplant-related investigations in adherence to WHO guidance for blood sampling in child health research.[15] In the case of a prolonged interval between the pretransplant blood sample and subsequent organ transplant (eg, >6 months) additional blood samples will be specifically requested from the patient/parent and verbal consent recorded in the medical record. Relevant clinical data will also be collected from participants at each study time point (online supplemental table 1).

Study investigations will consist of:

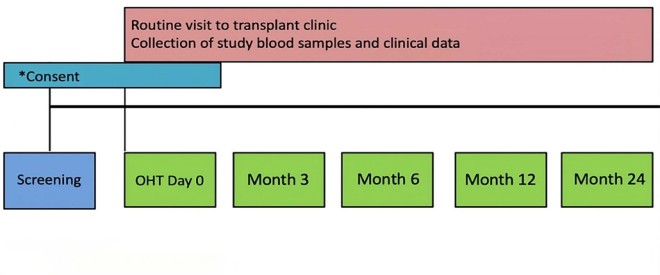

**Figure 1** Immunology of THymectomy And Childhood CArdiac transplant study flow chart. OHT, orthotopic heart transplant.

- ► Immunophenotyping of circulating immune cell subsets.
- ► EBV serology.
- ► EBV-specific T-lymphocyte quantification.
- ► EBV-specific T-lymphocyte functional analysis.

### Immunophenotyping of circulating immune cell subsets

Circulating immune cell populations will be analysed in Newcastle University laboratories. Mononuclear cells (MNCs) will be isolated and cryopreserved from blood samples taken at each study time point. In addition, biobanked MNC samples from age-matched healthy children will be included for analysis as non-thymectomy, non-transplant controls. Circulating immune cell subsets will be identified within samples using high-dimensional spectral flow cytometry (Aurora system (Cytek)). The 5-laser Aurora spectral cytometer enables an in-depth analysis of up to 40 cell surface markers at a time.[16] This is highly valuable in phenotypically characterising various immune cell populations and their subsets (including T-lymphocytes, B-lymphocytes, Natural Killer [NK] cells and dendritic cells) in low volume paediatric samples.[17] These samples will be analysed simultaneously in batches to mitigate interassay variation. A comprehensive list of cell surface markers has been selected to comprise the following:

1. Major innate/adaptive cell lineages.
2. Main T-lymphocyte subsets as well as putative T-helper subsets.
3. Recent thymic activity.
4. Cellular exhaustion and senescence.
5. Key innate/adaptive cells involved in immune response to EBV infection.

Two full spectrum flow panels have been validated for this purpose. A 24-colour panel has been designed to probe the wider circulating immune landscape of the study cohort (table 1). This is complemented by a targeted 30-colour T-lymphocyte panel that will elucidate any temporal changes in the T-lymphocyte compartment after transplantation (table 2).

### EBV serology

Routine blood samples for evaluation of EBV serology will be tested centrally in the Newcastle upon Tyne NHS Hospitals Foundation Trust (NUTH) virology laboratories. EBV and CMV viral load will be measured from whole blood using PCR assays. Serum and/or plasma will be tested for antibody response to key EBV proteins including IgM/IgG for viral capsid antigen and IgG for EBV Nuclear Antigen 1. The results for viral load assay will be reported as total titres while antibody testing will be reported as a binary detected/not detected based on NUTH laboratories reference cut-off values.

### EBV-specific T-lymphocyte quantification

An optimised panel of Human Leucocyte Antigen (HLA)-restricted peptide-Major Histocompatibility Complex (pMHC) I/II tetramers will be used to evaluate

   

**Table 1** Cell surface markers of interest used in the broad immune full spectrum flow for the Immunology of THymectomy And Childhood CArdiac transplant study

| Cell surface marker | Fluorophore | Antibody clone | Purpose |
|---|---|---|---|
| CD45 | Spark YG 593 | HI03 | Leucocytes |
| TCRγd | PerCP-eFluor 710 | B1.1 | γd T-lymphocyte |
| CD3 | BV 510 | SK7 | Pan T-cell, NK T-like cells |
| CD4 | PerCP | SK3 | CD4 T-helper lymphocytes |
| CD8 | BUV 805 | SK1 | CD8 cytotoxic T-lymphocytes |
| CD11c | PE-Cy7 | B-ly6 | Dendritic cell differentiation |
| CD14 | Spark Blue 550 | 63D3 | Monocyte differentiation |
| CD16 | BUV 496 | 3G8 | Monocyte, NK cell and dendritic cell differentiation |
| CD19 | Spark NIR 685 | HIB19 | B lymphocytes |
| CD20 | BV 786 | 2H7 | B lymphocytes |
| CD21 | PE-Cy5 | B-ly4 | B lymphocyte differentiation |
| CD24 | PE-AF 610 | SN3 | B lymphocyte differentiation |
| CD27 | APC-H7 | M-T271 | T/B lymphocyte differentiation |
| CD38 | APC-Fire 810 | HIT2 | Monocyte, dendritic cell, T/B lymphocyte activation and differentiation |
| CD56 | BUV 737 | NCAM16.2 | NK cells |
| CD57 | Pacific Blue | NK-1 | NK and CD8 T lymphocyte immune senescence |
| CD123 | Super Bright 436 | 6H6 | Plasmacytoid dendritic cells |
| CD127 | APC-R700 | HIL-7R-M21 | Cytokine receptor; T lymphocyte differentiation |
| KIR (CD158) | BUV 605 | DX27 | NK cells |
| NKG2A (CD159a) | APC | REA110 | NK cells |
| NKG2C (CD159c) | PE | REA205 | NK cells |
| IgD | BV 480 | IA6-2 | B lymphocyte differentiation |
| IgM | BV 570 | MHM-88 | B lymphocyte differentiation |
| HLA-DR | PE-Fire 810 | L243 | Monocyte activation, dendritic cell lineage, NK cell lineage discrimination, |
| Viability | Live/Dead Blue | – | Live cells |

NK, Natural Killer.

EBV-specific CD8 and CD4 T-lymphocyte immunity at the time points outlined in figure 1.[18] This panel will be tailored to maximise coverage of participants who have the most commonly expressed HLA genotypes within our study cohort. Combinations of differently fluorescently labelled pMHC tetramers presenting purified peptides of dominant lytic and latent EBV epitopes,[19] relevant for the patient HLA genotype, will be included in the T-lymphocyte panel where applicable (table 2). These customised tetramers will be procured from the National Institute of Health Tetramer Core Facility, Atlanta, Georgia, USA.

### EBV-specific T-lymphocyte functional analysis

To assess the functional capacity of EBV-specific T-lymphocytes in an HLA-unbiased manner, effector cytokine production of interferon gamma will be determined by ELISpot following stimulation with pools of overlapping peptides (JPT PepMix) representing the full sequences of a panel of EBV latent and lytic cycle proteins.[20 21]

### Data collection

Study data will be collated and managed using Research Electronic Data Capture (REDCap) electronic case report forms.[22] REDCap is a secure, web-based platform specifically designed to support data acquisition for research studies. It provides restricted user rights to protect identifiable data, including audit trails for tracking data handling and export and procedures for data integration with external sources.[23]

### Outcome measures

Primary outcomes measures:
1. Proportions of circulating innate and adaptive immune cell subsets before and at 3, 6, 12 and 24 months post-transplant.
2. Frequency of detectable EBV-specific T-lymphocyte immunity.
3. Functional capacity of EBV-specific T-lymphocytes.
   Secondary outcome measures:

**Table 2** Cell surface markers of interest used in the T lymphocyte full spectrum flow panel for the Immunology of THymectomy And Childhood CArdiac transplant study

| Cell surface marker | Fluorophore | Antibody clone | Purpose |
|---|---|---|---|
| CD45 | Spark YG 593 | HI03 | Leucocytes |
| CD45RA | BUV 395 | 5H9 | T lymphocyte differentiation |
| CD3 | BV510 | SK7 | Pan T-cell, NK T-like cells |
| CD4 | PerCP | SK3 | CD4 T-helper lymphocytes |
| CD8 | BUV 805 | SK1 | CD8 cytotoxic T lymphocytes |
| TCRγd | PerCP-eFluor 710 | B1.1 | γd T lymphocyte |
| CD25 | PE-AF700 | CD25-3G10 | Regulatory T lymphocytes |
| CD27 | APC-H7 | M-T271 | T lymphocyte differentiation |
| CD28 | BV650 | CD28.2 | T lymphocyte differentiation |
| CD31 | BV711 | WM59 | Recent thymic emigrants |
| CD38 | APC-Fire 810 | HIT2 | T lymphocyte activation/differentiation |
| CD39 | BUV 661 | TU66 | Regulatory T lymphocytes |
| CD57 | Pacific Blue | NK-1 | CD8 T lymphocyte immune senescence |
| CD62L | BUV 496 | SK-11 | T lymphocyte differentiation |
| CD69 | AF 647 | FN50 | T lymphocyte activation |
| CD95 | PE-Cy5 | DX2 | T lymphocyte activation/differentiation |
| CD127 | APC-R700 | HIL-7R-M21 | Chemokine receptor; T lymphocyte differentiation |
| HLA-DR | PE-Fire 810 | L243 | T lymphocyte activation |
| CCR4 | BB 700 | 1G1 | Chemokine receptor; T lymphocyte differentiation |
| CCR5 | BUV 563 | 2D7/CCR5 | Chemokine receptor; T lymphocyte differentiation |
| CCR6 | BV 480 | 140706 | Chemokine receptor; T lymphocyte differentiation |
| CCR7 | BV 421 | G043H7 | T lymphocyte differentiation |
| CXCR3 | AF 488 | GH025H7 | Chemokine receptor; T lymphocyte differentiation |
| CXCR5 | BV 750 | RF8B2 | Chemokine receptor; T lymphocyte differentiation |
| PD-1 | BV 785 | EH12.2H7 | Co-inhibitory receptor; T lymphocyte exhaustion |
| LAG-3 | PE-Cy7 | 3DS223H | Co-inhibitory receptor; T lymphocyte exhaustion |
| TIM-3 | BV 605 | F38-2E2 | Co-inhibitory receptor; T lymphocyte exhaustion |
| CTLA4 | PE-CF594 | BNI3 | Co-inhibitory receptor; T lymphocyte exhaustion |
| Tetramer (lytic) | APC | – | EBV-specific T lymphocytes |
| Tetramer (latent) | PE | – | EBV-specific T lymphocytes |
| Viability | Live/Dead Blue | – | Live cells |

EBV, Epstein-Barr virus; NK, Natural Killer.

1. Incidence of EBV infection.
2. Time from transplantation to EBV viraemia.
3. Time from EBV viraemia to seroconversion.

## DATA ANALYSIS PLAN
### Sample size calculation
As the study objectives are largely descriptive, no sample size calculation is necessary. Instead, we selected our sample size such that the study is feasible and large enough to conduct comprehensive analyses. Therefore, our sample size is based on a national average of 30 childhood cardiac transplants per year,[24] and a 60% recruitment rate. All eligible patients listed on the NHSBT register will be approached for consent, recognising that not all of these patients will receive a transplant during the lifetime of this study. However, any pretransplant blood samples that have been obtained are equally important to achieve the study's primary outcome measure.

### Statistical analysis
Data from spectral flow cytometry will be analysed using the OMIQ platform (https://www.omiq.ai/). Automated clustering and dimensionality reduction will be used to identify immune cell populations by FlowSOM and Uniform Manifold Approximation and Projection, respectively. These techniques overcome the practical

challenges associated with manual gating and user bias when analysing datasets from large flow panels.

Statistically different immune signatures between patients with early thymectomy and late or non-thymectomy, EBV+ and EBV– serostatus and between time points, will be identified by linear and generalised linear mixed models (statistical significance defined as $p < 0.05$). Continuous variables will be assessed by Pearson correlation using the single linkage method to group patients by expression values, and non-continuous variables by non-parametric Spearman correlation, as appropriate. Viral loads will be serially quantified at each study time point to correlate changes in immune responses with the volume of circulating virus-infected cells. Tetrameric frequencies for population groups (eg, early vs late/non-thymectomy) will be compared using Mann-Whitney U test and between time points for paired patient sample using Wilcoxon signed-rank test.

### Patient and public involvement

The Young Person's Advisory Group North England (YPAGne) was involved in the development of the study design, patient facing documents (eg, PIS) and the informed consent process. Ongoing consultation with YPAGne will continue to influence participant recruitment, outcome measure priorities and the acceptability of study methods.

### ETHICS

The ITHACA study has both research governance and ethical approvals (IRAS project ID: 298986; REC reference number: 21/NS/0142) and is adopted onto the National Institute for Health Research Clinical Research Network portfolio. Study organisation and sponsorship will be provided by The Newcastle upon Tyne Hospitals NHS Foundation Trust including coverage of insurance and NHS indemnity.

No significant risks are anticipated for enrolled participants. Study samples will be taken at the same time as routine transplant investigations during clinic visits, thereby avoiding any additional discomfort or hospital attendance. The estimated volume of study-related blood samples required at each assessment time point is based on guidance from the WHO for trial-related blood volumes in children.[15] This has been determined to be safe without any risk of causing anaemia.

### DISSEMINATION

Study findings will be disseminated widely through publications in high impact peer-reviewed journals, national and international conferences, and stakeholder events. We will engage YPAGne to identify patient-centred forums to facilitate discussion of study progress and any relevant findings with the general public. Published data will be made available via a public data repository, with a digital object identifier included in any published manuscript to aid discovery and outline access conditions. Potentially identifiable data, including patient sex, date of birth and date of transplant, will not be shared. Any unpublished data will only be shared with other parties where a data access agreement has been negotiated by Newcastle University Legal Services team on behalf of the study's chief investigator.

**Author affiliations**
[1]Wolfson Childhood Cancer Research Centre, Translational and Clinical Research Institute, Newcastle University Faculty of Medical Sciences, Newcastle upon Tyne, UK
[2]Department of Paediatric Haematology and Oncology, Great North Children's Hospital, Newcastle Upon Tyne, UK
[3]Department of Cardiothoracic Transplant, Great Ormond Street Hospital for Children NHS Foundation Trust, London, UK
[4]Department of Paediatric Nephrology, Great North Children's Hospital, Newcastle Upon Tyne, UK
[5]Department of Cardiopulmonary Transplantation, Newcastle Upon Tyne Hospitals NHS Foundation Trust, Newcastle Upon Tyne, UK
[6]Institute of Immunology and Immunotherapy, University of Birmingham, Birmingham, UK
[7]Translational and Clinical Research Institute, Newcastle University Faculty of Medical Sciences, Newcastle upon Tyne, UK
[8]Department of Paediatric Immunology and Haematopoietic Stem Cell Transplantation, Great North Children's Hospital, Newcastle Upon Tyne, UK
[9]Department of Cellular Pathology, Newcastle Upon Tyne Hospitals NHS Foundation Trust, Newcastle Upon Tyne, UK

**Acknowledgements** The authors thank the study participants and their families. We acknowledge the significant contribution of the YPAGne team in refining the ITHACA study design. We would also like to thank Lesley Brindley (clinical trials co-ordinator) and all of the transplant co-ordinators and transplant nurse specialists for their assistance in the screening and recruitment of study participants.

**Contributors** UTO and SB conceptualised the study. UTO, SB, CMB, ARG and HML contributed to the study design. SB (chief investigator) and coapplicants UTO, CMB and ARG developed the protocol with contributions from PH, MO, GP, AK, JS and ZR. UTO drafted and prepared the study protocol manuscript and received comments from the coauthors. All authors reviewed and approved the final version of the manuscript for publication.

**Funding** This study is funded by the Lymphoma Research Trust (N/A) and Cancer Research UK (PHSTU-Hist\100297). SB is funded by an MRC Clinician Scientist Fellowship (MR/S021590/1).

**Competing interests** None declared.

**Patient and public involvement** Patients and/or the public were involved in the design, or conduct, or reporting, or dissemination plans of this research. Refer to the Data analysis plan section for further details.

**Patient consent for publication** Not applicable.

**Provenance and peer review** Not commissioned; externally peer reviewed.

**ORCID iD**
Ugonna T Offor http://orcid.org/0000-0002-7207-3175

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
