## [Reviewer comments · BMJ Open]

ARTICLE DETAILS

TITLE (PROVISIONAL)	The Immunology of Thymectomy And childhood CArdiac transplant (ITHACA): Protocol for a UK-wide prospective observational cohort study to identify immunological risk factors of post-transplant lymphoproliferative disease (PTLD) in thymectomised children.
AUTHORS	Offor, Ugonna; Hollis, Paolo; Ognjanovic, Milos; Parry, Gareth; Khushnood, Abbas; Long, Heather; Gennery, Andrew; Bacon, Chris; Simmonds, Jacob; Reinhardt, Zdenka; Bomken, Simon

VERSION 1 – REVIEW

REVIEWER	Olov Ekwall University of Gothenburg, Dept Pediatrics
REVIEW RETURNED	22-Sep-2023

GENERAL COMMENTS	The manuscript describes a well-designed study addressing an important clinical question. The study is clearly described and scientifically sound. One suggestion for the authors is, if still possible, to consider adding one more control group consisting of thymectomized children that have not undergone heart transplantation. This would add a possibility to disentangle the effects of thymectomy from effects caused by immunosuppression and allotransplantation. Note that this is just a suggestion, and not to be considered as a revision needed for the publication of the protocol. Minor comments: It is ambitious, and at the same time challenging, to assess EBV-specific T-lymphocytes from all study participants using tetramers considering the need to match tetramers to the individual HLA-haplotypes. Please clarify if this analysis will be done for all participants, or only for those with the more common HLA-haplotypes. Cell surface markers used for immune phenotyping are not presented in detail which makes it hard to assess this part of the study. It would be of value to add detailed information on markers, and if possible, also on which specific antibody clones that are used.
--

REVIEWER	Diana M. Metes UPMC, Surgery
REVIEW RETURNED	22-Sep-2023

GENERAL COMMENTS	This is a study protocol to investigate the immunology of thymectomy after cardiac transplantation to identify immunologic
--

	risks for PTLT in thymectomised children. This is a well written and important study. I have a few comments that need to be addressed before the protocol is acceptable for publication.  1. The authors should clarify whether this is ongoing or due to start. They should indicate the dates of the study 2. Inclusion criteria: a) should consider only EBV- seronegative serostatus of the recipient since historically only EBV- seronegative recipients at Tx were candidates at risk for PTLT. b) I would only keep cadaveric KTx recipients for consistency with heart Tx recipients (organ living donor recipients are doing better immunologically then cadaveric recipients). 3. Assessment and procedures: Important to document if the Tx recipient undergo:  a) First surgery for congenital heart disease and subsequently heart Tx or directly heart Tx. b) total of partial thymectomy, since these could have a different influence on the T cell output. c) Please include these in Supplementary table 1 under I baseline patient details.
--	--

VERSION 1 – AUTHOR RESPONSE

Reviewer #1: The manuscript describes a well-designed study addressing an important clinical question. The study is clearly described and scientifically sound.

We are grateful to reviewer 1 for acknowledging the clarity and scientific robustness of our study protocol.

One suggestion for the authors is, if still possible, to consider adding one more control group consisting of thymectomized children that have not undergone heart transplantation. This would add a possibility to disentangle the effects of thymectomy from effects caused by immunosuppression and allotransplantation. Note that this is just a suggestion, and not to be considered as a revision needed for the publication of the protocol.

We thank reviewer 1 for this suggestion. The inclusion of a thymectomised non-transplant control group was a point that was strongly considered during our study design phase. However, we are cognisant that the immunological impact of early thymectomy on non-transplanted children has already been investigated and reported extensively, including by *Prelog et al (2009)*, *Sauce et al (2009)*, *Halnon et al (2005)*, *Mancebo et al (2008)*, *Madhok et al (2005)* and *Morsheimer et al (2016)*. Collectively, these studies have examined the long-term impact of childhood thymectomy on the frequency and phenotypes of many of the immune cell subsets that are of interest to our study, including those that play a pivotal role in EBV response.

To mitigate any confuscation of the effects of thymectomy and immunosuppression, we have obtained the datasets, including raw flow cytometry files where possible, for any studies that has these available in public repositories. This will allow us to include them as a comparator during our data analysis. Furthermore, we plan to use pre-transplant blood samples from study participants who have undergone early thymectomy during initial palliative surgery to parse any confounding impact(s) of the two procedures. To further improve the robustness of our analysis, we have included biobanked

mononuclear cell samples from age-matched healthy children as part of our immunophenotyping assay. This will allow us to establish the peripheral immune landscape in non-thymectomised, non-transplanted children as an additional comparator. We have amended the methods section of the manuscript under “Immunophenotyping of immune cell subsets” to reflect the inclusion of this control group.

Minor comments:

It is ambitious, and at the same time challenging, to assess EBV-specific T-lymphocytes from all study participants using tetramers considering the need to match tetramers to the individual HLA-haplotypes. Please clarify if this analysis will be done for all participants, or only for those with the more common HLA-haplotypes.

We have updated the methods section under “EBV-specific T-lymphocyte quantification” to highlight that the inclusion of tetramers in our flow panel will be limited to only the participants with the most commonly expressed HLA-genotypes in our study cohort. However, it is important to note that of the 32 patients for whom we already have HLA-haplotype data, our tetramer panel will currently provide coverage across 88% of patients.

Cell surface markers used for immune phenotyping are not presented in detail which makes it hard to assess this part of the study. It would be of value to add detailed information on markers, and if possible, also on which specific antibody clones that are used.

We apologise for this omission. We have included relevant text in the methods section under “Immunophenotyping of immune cell subsets” to give further details about the flow panels we will use for immune phenotyping. We have also included two main tables (Tables 1 and 2) that outline the cell surface markers, specific antibody clones and the fluorophores used.

Reviewer #2: This is a study protocol to investigate the immunology of thymectomy after cardiac transplantation to identify immunologic risks for PTLD in thymectomised children. This is a well written and important study.

We thank reviewer 2 for highlighting the value of our study.

I have a few comments that need to be addressed before the protocol is acceptable for publication.

1. The authors should clarify whether this is ongoing or due to start. They should indicate the dates of the study.

We apologise for not including this important detail. We have updated our methods section under “study design and setting” to specify that our study opened to patient recruitment in March 2022 and is planned to close in June 2024.

2. Inclusion criteria: a) should consider only EBV- seronegative serostatus of the recipient since historically only EBV- seronegative recipients at Tx were candidates at risk for PTLD.

We agree that there is a well-established link between primary EBV infection and the risk of PTLD. Additionally, however, there is good evidence that some children who are EBV seropositive at transplant are also affected by PTLD, albeit to a lesser degree. A recent systematic review by *Haider et al* (2020) has shown that roughly 23% of paediatric renal transplant patients who develop PTLD are EBV sero-positive at the time of transplant. Similarly, within the paediatric heart transplant population, our previous study *Offor et al* (2021) highlighted that 13 out of 35 patients who developed PTLD were EBV positive prior to transplantation. The phenotypic and functional similarities/differences in immune risk profiles required for EBV-driven PTLD and their relationship to pre-transplant serostatus has not been categorised. We therefore believe that our study hypothesis can only be addressed with the appropriate scientific rigour if we include both EBV naïve and experienced patients in our study cohort.

b) I would only keep cadaveric KTx recipients for consistency with heart Tx recipients (organ living donor recipients are doing better immunologically than cadaveric recipients).

It is true that living donor kidney recipients have better immunological outcomes than those transplanted with a cadaveric kidney. However, there is conflicting evidence about how this might affect the risk of PTLD. While *Dharnidharka et al* (2001) reported a significant association between cadaveric renal transplant in children and the incidence of PTLD, this finding has not been replicated in more recent studies by *Francis et al* (2018) and *Hyun et al* (2019). Factors such as the improvement to immunosuppression protocols within the past 10-15 years may have contributed to this change in risk profile. We have also identified that an overwhelming majority of patients currently listed for a kidney transplant at our centre are awaiting living donor transplantation. This in itself is compounded by the rarity of paediatric kidney transplants. Limiting our study inclusion to only cadaveric kidney transplants is likely to have a significant impact on our ability to successfully recruit patients within the study's timeframe.

Of relevance to both points 2a and 2b above, we have clarified that the ITHACA study is now actively recruiting and indeed over halfway to full recruitment. We would not consider it ethically appropriate to retrospectively exclude patients who have been recruited. Furthermore, with the balance of patients recruited to date we believe we are fully able to address the stated trial objectives.

3. Assessment and procedures: Important to document if the Tx recipient undergo:

a) First surgery for congenital heart disease and subsequently heart Tx or directly heart Tx.

- b) total of partial thymectomy, since these could have a different influence on the T cell output.
- c) Please include these in Supplementary table 1 under I baseline patient details.

Our case report forms have been designed to capture the date and type of first major cardiac surgery (prior to transplant) for patients with congenital heart disease or acquired cardiomyopathy where relevant. This also includes data on the type of thymectomy (total vs partial). We note that this wasn't made clear in supplementary table 1 and have now updated it with the relevant details.

VERSION 2 – REVIEW

REVIEWER	Olov Ekwall University of Gothenburg, Dept Pediatrics
REVIEW RETURNED	05-Oct-2023
GENERAL COMMENTS	The minor comments I raised in my initial review has been met in the revised version of the manuscript.
REVIEWER	Diana M. Metes UPMC, Surgery
REVIEW RETURNED	09-Oct-2023
GENERAL COMMENTS	The authors have addressed my comments.